ecology

conservation funds, extinction risk, IUCN, Natura 2000, online popularity, taxonomic bias

**Author for correspondence:**
Stefano Mammola
e-mail: stefano.mammola@helsinki.fi;
stefano.mammola@cnr.it

# Towards a taxonomically unbiased European Union biodiversity strategy for 2030

Stefano Mammola[1,4], Nicoletta Riccardi[4], Vincent Prié[5], Ricardo Correia[2,3,6,7], Pedro Cardoso[1], Manuel Lopes-Lima[8] and Ronaldo Sousa[9]

[1]Laboratory for Integrative Biodiversity Research (LIBRe), Finnish Museum of Natural History (LUOMUS),
[2]Helsinki Lab of Interdisciplinary Conservation Science (HELICS), Department of Geosciences and Geography, and
[3]Helsinki Institute of Sustainability Science (HELSUS), Department of Geosciences and Geography, University of Helsinki, 00100 Helsinki, Finland
[4]Water Research Institute (IRSA), National Research Council (CNR), 28922 Verbania Pallanza, Italy
[5]Institute of Systematics, Evolution, Biodiversity (ISYEB), National Museum of Natural History (MNHN), CNRS, SU, EPHE, UA, CP 51, 75005 Paris, France
[6]DBIO and CESAM—Centre for Environmental and Marine Studies, University of Aveiro, 3810-193 Aveiro, Portugal
[7]Instituto de Ciências Biológicas e da Saúde, Universidade Federal de Alagoas, Campus A. C. Simões, Avenida Lourival Melo Mota, Tabuleiro dos Martins, Maceió, 57072-900 Alagoas, Brazil
[8]CIBIO/InBIO—Research Center in Biodiversity and Genetic Resources, University of Porto, Campus Agrário de Vairão, 4485-661 Vairão, Portugal
[9]CBMA—Centre of Molecular and Environmental Biology, Department of Biology, University of Minho, Campos de Gualtar, 4710-057 Braga, Portugal

(iD) SM, 0000-0002-4471-9055; RC, 0000-0001-7359-9091; RS, 0000-0002-5961-5515

Through the Habitats Directive (92/43/EEC) and the financial investments of the LIFE projects, Europe has become an experimental arena for biological conservation. With an estimated annual budget of €20 billion, the EU Biodiversity Strategy for 2030 has set an ambitious goal of classifying 30% of its land and sea territory as Protected Areas and ensuring no deterioration in conservation trends and the status of protected species. We analysed LIFE projects focused on animals from 1992 to 2018 and found that investment in vertebrates was six times higher than that for invertebrates (€970 versus €150 million), with birds and mammals alone accounting for 72% of species and 75% of the total budget. In relative terms, investment per species towards vertebrates has been 468 times higher than that for invertebrates. Using a trait-based approach, we show that conservation effort is primarily explained by species' popularity rather than extinction risk or body size. Therefore, we propose a roadmap to achieve unbiased conservation targets for 2030 and beyond.

## 1. Introduction

Overwhelming evidence exists that most Earth ecosystem processes are being altered by human activities, suggesting that we may have entered a human-dominated geological epoch—the 'Anthropocene' [1]. It is largely accepted that humans are causing the sixth mass species extinction [2], which can be considered a clarion call to increase global efforts to study, halt, and possibly reverse the ongoing negative environmental trends. Europe is no exception, given that it has a long experience of human disturbance and consequent biodiversity loss [3]. At the same time, since the Habitats Directive (92/43/EEC) was established in 1992, the European Union (EU) has acted as a global test case for practical conservation and restoration of natural habitats and their wild flora and fauna.

Although the Habitats Directive and the parallel financial investment on LIFE projects—the EU flagship funding instrument for the environment and climate

action created in 1992—are seen as pioneer endeavours and represent a strong legal and financial tool for biodiversity conservation, most species in Europe continue to decline [3]. However, evidence suggests that when enough resources are invested in species-level conservation, we are able to halt these negative trends. In Europe, for example, there are positive rewilding trends in a few charismatic large carnivores that have received constant conservation funding through LIFE projects [4]. In the same vein, examples of effective allocations of funds to species conservation come from Australian threatened birds [5] and North America birds inhabiting wetlands [6].

At the end of the trial period of the Habitats Directive, the EU launched a new Biodiversity Strategy intended to create Protected Areas for 30% of its land and sea territory by 2030 and ensure no deterioration in conservation trends and the status of protected species and habitats [7]. The ideals of this ambitious plan resonate with that of other similar projects such as the Global Deal for Nature [8] or the Half-Earth project (E.O. Wilson Biodiversity Foundation), which aims to protect up to 50% of the Earth's ecoregions. Such an effort will be critical if we are to embrace the recent proposal of keeping known species extinctions to below 20 a year over the next 100 years [9].

These conservation schemes should ideally encompass all native ecosystem types, species, and ecological successions, in order to ensure not to 'add more land to reach [a] global target that is similar to what is already well accounted for at the expense of underrepresented habitats and species' [8]. It is, therefore, essential to assess whether conservation investment is optimally allocated among species and habitats, or whether taxonomic and other biases still permeate biological conservation [10–12]. For example, Moser et al. [13] showed that over the 26 years of LIFE projects, strong disagreements have arisen over the Red Lists and the Habitats Directive protected species list, suggesting the need for a careful revision of EU policy. As a result, important initiatives have been put forward, most notably the recent proposal by the EU to account more comprehensively for invertebrates in the LIFE program [14].

With the new Biodiversity Strategy for 2030, in addition to the economic recovery plan in response to COVID-19 highly focused on the biodiversity and climate crises, the EU is proposing a large annual budget for spending on nature (an estimated €20 billion [7]). Among other uses, this budget will finance direct conservation actions towards protected species. Therefore, the time is ripe to assess the achievements of the Habitats Directive and LIFE projects in the past and look forward to the goal of establishing an unbiased conservation agenda in Europe. To quantify long-term taxonomic biases and potential drivers and their possible impact on conservation strategies and decision-making policies, we mined information on LIFE projects conducted between 1992 and 2018 that were focused on animals ($n = 835$). Here, our aim was to obtain a comprehensive picture of the number of applied conservation initiatives and allocation of the LIFE's conservation budget across the animal tree of life in Europe.

## 2. Methods

### (a) Extraction of data from the LIFE project

We extracted information on the amount of funding allocated to various species using the LIFE projects database (https://ec. europa.eu; accessed between February and May 2020). Note that we focused here on a species-level conservation approach [9], in contrast with other more general conservation measures (such as socio-ecological approaches [15,16] and others [17]) that are only indirectly covered by LIFE projects. We first filtered LIFE projects specifically aimed at species conservation, using the query STRAND = 'All'; YEAR = 'All'; COUNTRY = 'All'; THEMES = ' Species'; SUB-THEMES = 'Amphibians'; 'Birds'; 'Fish'; 'Invertebrates'; 'Mammals'; 'Reptiles'. This query resulted in 819 LIFE projects that met the search criteria. A second query focused on THEMES = 'Biodiversity issues'; SUB-THEMES = 'Ecological coherence'; 'Invasive species'; 'Urban biodiversity', which matched an additional 298 LIFE projects. For the latter query, we examined summaries of the LIFE projects and extracted further information only from those specifically aimed at the conservation of animal species ($n = 16$). For example, projects aimed at generically enhancing biodiversity through measures targeting ecosystems or the impacts of anthropic activities were not considered. In total, we included 835 projects in our analyses—819 with the theme 'Species' and 16 with the theme 'Biodiversity issues'. To define the amount of funds allocated to each species for each LIFE project, the budget of each project with multiple species was divided equally among the target species.

### (b) Calculation of species traits

To obtain a deeper understanding of the factors underlying the observed pattern of conservation measures among species, we investigated whether the number of LIFE projects and the budget allocation for each species was driven by its risk of extinction, body size, and/or online popularity.

We estimated the extinction risk category of each species in our database using the International Union for Conservation of Nature Red List of Threatened Species [18]. Using application programming interface (API) keys, we automatically assigned each species to its IUCN extinction risk category, and then we manually checked this matching to correct potential mistakes.

Body size is one of the few traits that is fully comparable across distant taxa [19], and it is also one of the most conspicuous traits correlating with extinction risk [20]. We estimated body size for birds using the Collins Bird Guide [21], for mammals using Aulagnier et al. [22], for reptiles and amphibians using Speybroek et al. [23], and for fish using FishBase [24]. For invertebrates, we estimated body size using diverse sources, from Wikipedia to original species descriptions and expert opinion and unpublished data by the authors (e.g. molluscs and insects).

We characterized the online popularity of each species in our database using a culturomics approach [25] based on the volume of Internet searches performed on Google's search engine. We obtained data on the average monthly relative search volume recorded between January 2010 and December 2019 for each species from the Google Trends API. Google Trends returns relative search volume data ranging from 0 to 100; the maximum value is assigned to the highest proportion of total searches observed during any given month of the sampled period and all other values are scaled in relation to it. To ensure comparable data between species, we collected the data using an approach similar to that described by Davies et al. [26]. We performed topic searches for combinations of multiple species, validating each species-specific topic beforehand [27], and ensuring one common species between each search. The values returned for this species in either search were used to estimate a scaling factor between searches, calculated as the coefficient of a linear regression between the monthly values of either search. The species used to calculate the scaling factor were selected iteratively to ensure the scaling factor was calculated as accurately as possible based on (i) the highest number of non-zero values between searches and (ii) a regression $R^2$ value above 0.95. The monthly values of search interest for each species were then rescaled using this coefficient to ensure estimates were

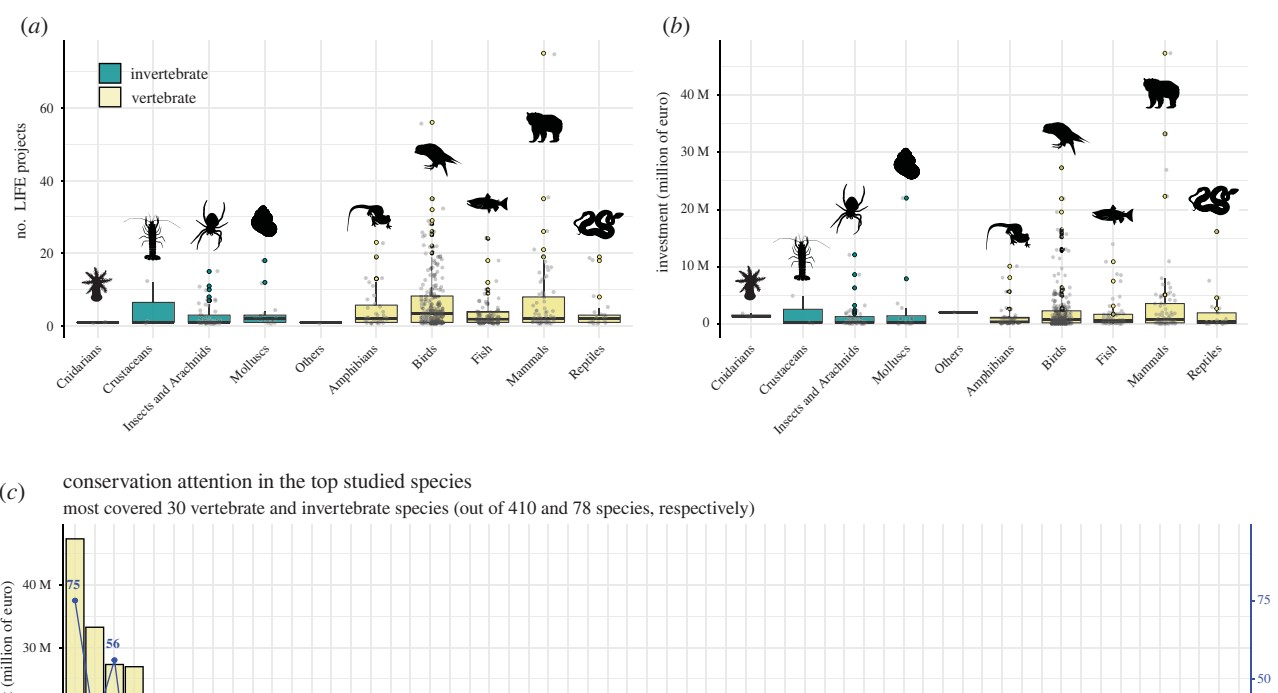

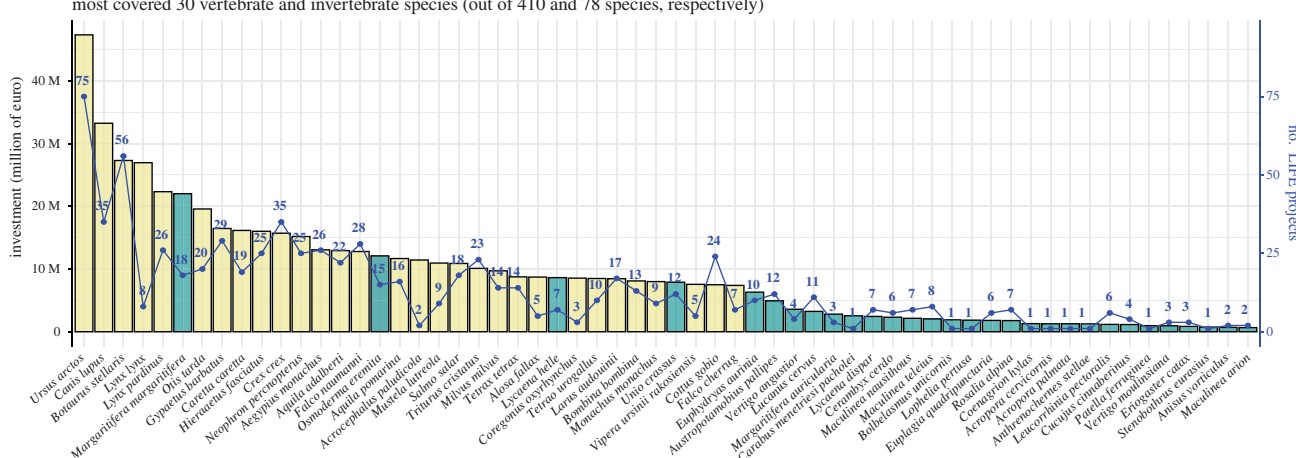

**Figure 1.** Breakdown of the number of projects (*a*) and budget allocation (*b*) across main animal groups covered by the LIFE projects (*n* = 835). (*c*) The most covered 30 species of vertebrates (out of 410) and invertebrates (out of 78) in the LIFE projects analysed (*n* = 835). The vertical bar represents monetary investment and the blue scatter line the number of LIFE projects devoted to each species. (Online version in colour.)

comparable between species and averaged across 10 years of data (i.e. 120 months of search volume). The resulting metric provides an estimate of the average frequency with which each species was searched for every month over the last 10 years relative to the other species in our data. This metric can be seen as a proxy for the underlying degree of public interest in each species over the sampled time period [28].

## (c) Statistical analyses

We explored the relationships between the three traits and conservation attention with a generalized linear mixed model that accounted for taxonomic non-independence among species [29].

We performed the analyses in R [30], initially exploring the dataset following a general protocol for data exploration [31]. We checked the homogeneity of continuous variables and log-transformed non-homogeneous variables, when appropriate. We verified multicollinearity among predictors with pairwise Pearson's *r* correlations (electronic supplementary material, figure S1), setting the threshold for collinearity at | *r* | > 0.7 [31]. We visualized potential associations between continuous and categorical variables with boxplots (electronic supplementary material, figure S2). Given that average online popularity (logarithm) and relative popularity (logarithm) were highly correlated (*r* = 0.96), we included only average popularity in the regression analysis. Furthermore, since the number of LIFE projects and total budget were highly correlated (*r* = 0.79), we expressed conservation attention only as the number of LIFE projects (dependent variable).

We fitted a generalized linear mixed model to the data with the R package *lme4* [32]. The mixed part of the model allowed us to

take into account taxonomic non-independence of observations, under the assumption that closely related species should share more similar traits than would be expected of a random sample of species. More specifically, the taxonomic relatedness among species was accounted for with a nested random intercept structure (Class/Order/Family). Given that the response variable number of projects are counts, we initially selected a Poisson error and a log link function. The Poisson model was, however, slightly over-dispersed (dispersion ratio = 3.5; Pearson's $\chi^2$ = 1463.5; *p*-value < 0.001), and thus we switched to a negative binomial distribution. Model validation was performed with the aid of the R package *performance* [33].

## 3. Results and discussion

## (a) Allocation of projects and fundings

We found that the number of LIFE projects (figure 1*a*) and budget allocation (figure 1*b*) varied substantially across faunal groups. Overall, LIFE projects focused on 410 vertebrate and 78 invertebrate species. Net monetary investment for vertebrates was over six times higher than for invertebrates (approx. €970 versus €150 million; electronic supplementary material, table S1), a long-known taxonomic bias [10] that seemingly persists in conservation throughout Europe [34]. This bias in conservation efforts is even more striking when it is relativized to the actual numbers of known species in Europe, namely around 1800 species of vertebrates and 130 300 species of invertebrates [35]. In relative terms, 23% of vertebrates

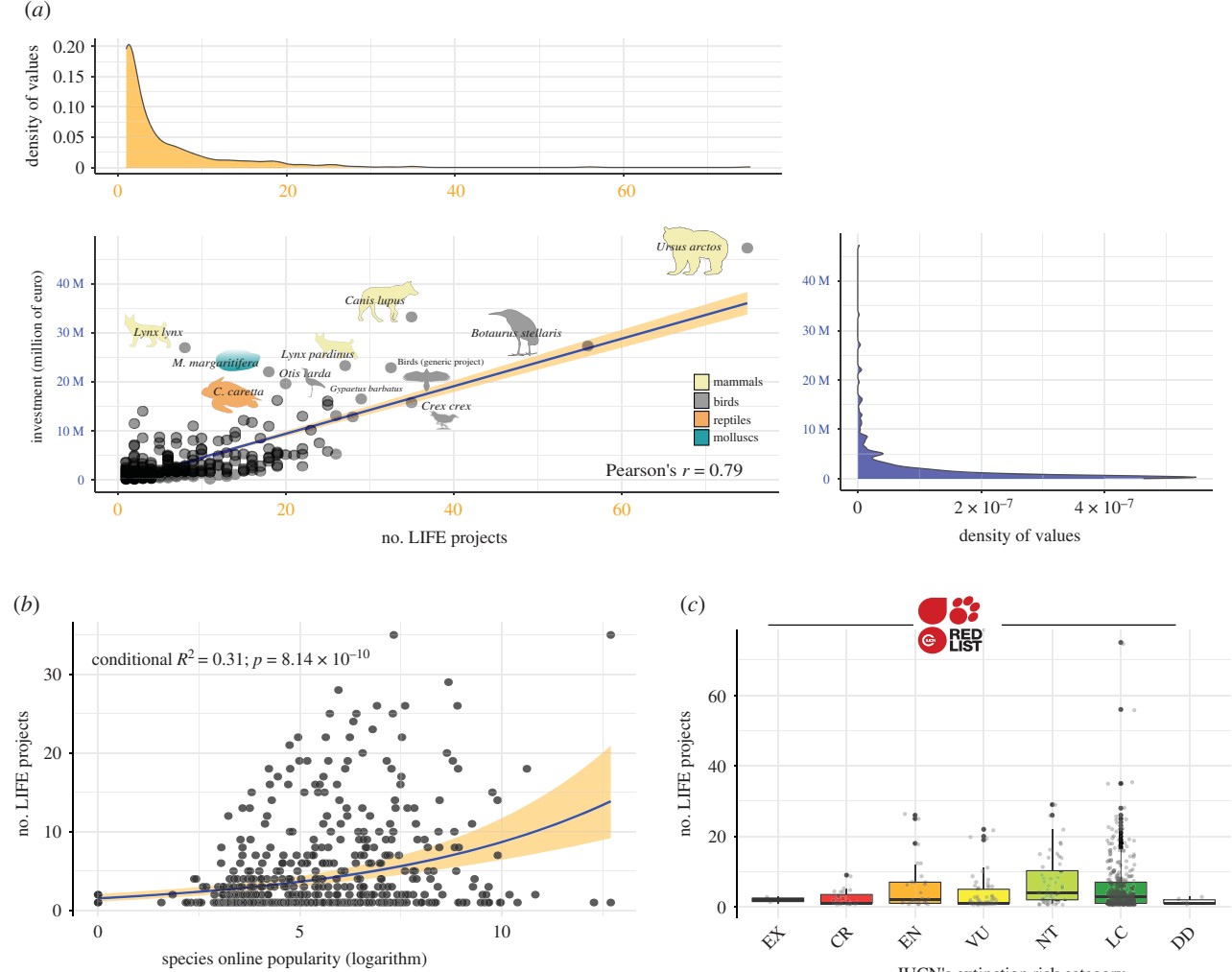

**Figure 2.** (a) Pearson's r correlation between monetary investment and the number of projects for the 78 invertebrate and 410 vertebrate species covered by the LIFE projects (n = 835). The farther away a dot is from the correlation line, the more the conservation effort is unbalanced towards either the number of projects or the monetary investment. (b) Predicted relationship (blue line) and 95% confidence interval (orange shaded surface) between the number of LIFE projects and the species' online popularity, according to the results of a negative binomial generalized linear mixed model (electronic supplementary material, table S2). Online popularity is measured as the net online attention each species receives. (c) Breakdown of LIFE projects according to the IUCN extinction risk of the species that they cover (EX, Extinct; CR, Critically Endangered; EN, Endangered; VU, Vulnerable; NT, Near Threatened; LC, Least Concern; DD, Data Deficient). (Online version in colour.)

received funding versus only 0.06% of invertebrates, and the total investment per species for vertebrates was 468 times higher than for invertebrates. Not counting the species of invertebrates as yet to be discovered [36], including in Europe.

The top 30 invertebrate species, in terms of LIFE funds allocation, were coleopterans, butterflies, dragonflies, and a few molluscs (figure 1c); but this funding is miniscule, given invertebrate diversity and the current rate of declines across diverse habitats [37–39]. Even within invertebrates, the biases are notorious, with widespread, large, and/or colourful species being dominant [40]. Among vertebrates, 54% of species covered by LIFE projects were birds (accounting for 46% of the budget allocated) and 18% mammals (24% of the budget), with only 8 out of the 30 most protected vertebrate species not belonging to these two groups (figure 1c).

As expected, we found a significant positive correlation between budget allocation and the number of projects (figure 2a), although with key differences in conservation strategies for distinct species. Whereas the majority of species were covered by a small number of projects with a limited budget (€0–5 million), some outlying species were the target of more

intense budget allocation and a higher number of projects (e.g. the bear and great bittern), or a lower number of high-budget projects (e.g. the lynx and wolf). The unique outlier among invertebrates was the freshwater pearl mussel *Margaritifera margaritifera*, which was the focus of a small number of high-budget projects (figures 1c and 2a). It should be noted, however, that all these species are distributed over broad geographical expanses encompassing several EU countries and thus are more likely to be targeted by researchers applying to LIFE projects. Ironically, these broad geographical ranges possibly make these species less prone to extinction.

## (b) Drivers of taxonomic bias

We found that the only significant predictor explaining the conservation attention a species receives is online popularity (negative binomial GLMM: estimated $\beta \pm$ s.e.: $0.19 \pm 0.03$, $z = 6.14$, $p < 0.001$; electronic supplementary material, table S2). In particular, the species covered by a greater number of LIFE projects were also those which attracted the most interest online (figure 2b), suggesting that conservation in

the EU is largely driven by species charisma, rather than objective features. This result aligns with a recent study documenting a similar trend in a global sample of threatened vertebrates [26].

Unexpectedly, risk of extinction did not seem to affect LIFE fund allocation, as the species-under-extinction-threat categories failed to receive significantly higher conservation attention (see [41]). On the contrary, we found that the majority of projects focused on Least Concern and Near Threatened species (figure 2c). Whereas some of these non-threatened species may be broadly distributed, thereby acting as umbrella species for the habitat that they occupy and the species they coexist with, this investment bias seems unjustified given the limited budget of past LIFE projects. However, it is also worth noting that the use of umbrella species to protect other faunal groups has given mixed results with several examples of poor surrogacy [42,43].

The non-significant effect of body size was also somewhat unexpected, as this trait is often a good proxy of species' extinction risk [20,44], although it depends on the taxon [45]. However, it should be noted that most invertebrates in the Habitats Directive are large species. Moreover, the correlation between online popularity and body size was quite high (electronic supplementary material, figure S3), and thus the strong and significant effect of popularity may have partly masked a weaker effect of body size in explaining conservation attention.

## (c) A way forward

The EU's commitment to the ambitious goals of the Biodiversity Strategy for 2030 appears evident when considering the proposed financial plan for nature conservation [7]. At least €20 billion a year will be unlocked for spending on nature, which will require mobilizing private and public funding at a national and EU level, through a range of different programmes. Moreover, as nature restoration will make a major contribution to climate change mitigation objectives, a significant proportion of the 25% of the EU budget dedicated to climate action will be invested in biodiversity and nature-based solutions. All these actions go hand-in-hand with the EU Green Deal, which emphasizes the post-COVID-19 economic recovery, with the intention to invest further in conservation and sustainable development. However, based on past investment and taking the Habitats Directive and LIFE projects as a proxy, it seems likely that a few charismatic species will receive almost all the attention in the context of species-based conservation funding. When striving to assign the status of protected area to 30% of the EU's territory and halt the documented trends in species extinction [9], especially the silent extinctions of invertebrates [38], it is essential to overcome the prevailing taxonomic [34] and other [46] biases. We, therefore, propose a roadmap for more equitable species-focused conservation investments in the next decade.

First, we should promote species inventories and data compilation, to overcome the main knowledge gaps [47,48]. Aside from funding, overcoming such gaps will require creativity in using diverse sources of data, including monitoring schemes that already exist at the national (e.g. recording and monitoring schemes in the UK and Germany) and EU levels (e.g. Water Framework Directive monitoring for freshwater taxa), citizen science projects [49], and Internet-derived data (e.g. through *iEcology*; [50]).

Second, armed with such knowledge, we should quickly assess the status of a broader sample of species to obtain a real picture of the level of diversity that is threatened by human activities. Currently, only a handful of invertebrates are included in the IUCN Red List, preventing us from fully understanding their status, trends, and conservation needs [40].

Third, it is critical to review the Habitats Directive Annexes and thereby produce unbiased criteria for species protection [48]. This will allow us to build sound scenarios for the possible enlargement of the Natura 2000 network to implement the 30% target for protected areas in EU Member States [3].

Fourth, we should focus on endangered species and their habitats rather than directing substantial conservation efforts towards non-threatened species (figure 2c). The EU Biodiversity Strategy for 2030 will request the Member States to ensure no deterioration in conservation trends and the status of all protected habitats and species by 2030. This goal will be reached only by directing conservation efforts to species at greater risk of extinction, and by focusing on those taxa and habitats that are currently not accounted for [8].

Fifth, going forward, it will be necessary to monitor conservation priorities and funding investment on an annual basis. The EU currently lacks a comprehensive governance framework for steering the implementation of biodiversity commitments agreed at the national, European, or international level. To address this gap, the Commission is planning a new European biodiversity governance framework to map obligations and commitments and set out a roadmap to guide their implementation. As part of this new framework, a monitoring and review mechanism will be established, including a clear set of agreed indicators. It is within this framework that we must challenge the current taxonomic bias and establish a more equitable redistribution of funds.

The EU has recently released a technical report making clear the intent to more comprehensively account for invertebrates in the LIFE program [14]. Considering the number of European species in question [35], we realize that it will be impossible to implement this agenda for all invertebrate species and habitats [51,52]. As proposed by Cardoso & Leather [53], the simplest solution will be to maximize phylogenetic and functional coverage of the species targeted by the LIFE projects, by applying a positive discrimination mechanism during their evaluation. This can be achieved by including a taxonomic component in future project assessments and weighting its score according to objective criteria based on phylogenetic, functional, and spatial uniqueness in relation to previous projects. In this way, species that have received substantial funding in the past (e.g. the bear and lynx) would see their scores down-weighted; conversely, the project score of species that have never received funds (e.g. most invertebrates) would increase. These, or similar approaches and criteria, will increase the objectivity of the EU's future conservation planning, allowing it to lead the world by example and action.

**Data accessibility.** Data supporting this study and R code to generate graphs and analyses are available in the Dryad Digital Repository: https://dx.doi.org/10.5061/dryad.1rn8pk0s0 [54].

**Authors' contributions.** M.L.-L., N.R., R.S., S.M., and V.P. conceived the idea. N.R. mined data from LIFE projects. R.C. extracted data from IUCN and the cultural value of species. N.R., R.S., and S.M. extracted species body size from different sources. S.M. performed analyses

and prepared figures. R.S. and S.M. wrote the first draft of the paper. All authors critically contributed to the writing of the paper through comment, additions, and revisions.

Competing interests. We declare we have no competing interests.

Funding. R.C. is funded by the Helsinki Institute of Sustainability Science (HELSUS) and the University of Helsinki. S.M. is supported by the European Union's Horizon 2020 research and innovation programme under grant agreement no. 882221.

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
