## [Reviewer comments · Proceedings of the Royal Society B: Biological Sciences]

Review History

RSPB-2020-2166.R0 (Original submission)

Review form: Reviewer 1

Recommendation

Accept with minor revision (please list in comments)

Scientific importance: Is the manuscript an original and important contribution to its field?

Excellent

General interest: Is the paper of sufficient general interest?

Excellent

Quality of the paper: Is the overall quality of the paper suitable?

Excellent

Is the length of the paper justified?

Yes

Should the paper be seen by a specialist statistical reviewer?

No

Do you have any concerns about statistical analyses in this paper? If so, please specify them explicitly in your report.

No

It is a condition of publication that authors make their supporting data, code and materials available - either as supplementary material or hosted in an external repository. Please rate, if applicable, the supporting data on the following criteria.

Is it accessible?

Yes

Is it clear?

Yes

Is it adequate?

Yes

Do you have any ethical concerns with this paper?

No

Comments to the Author

It is well known that there is a bias in the funds allocated to the conservation of species with a clear positive bias towards vertebrates. Invertebrates were clearly overlooked. This manuscript reports on an interesting evaluation of the taxonomic biases on LIFE projects conducted between 1992 and 2018, and potential drivers and their possible impact on conservation strategies and decision-making policies. The research was well conducted and I only suggest some minor improvements.

Some comments for improvement:

-In the Introduction, you should also mention that LIFE program is now targeting LIFE projects focusing on Invertebrates and refer to:

Neemo (2020) LIFE and Invertebrates-Stepping up to the challenges: Conclusions of the 2018 LIFE Platform Meeting on invertebrates and 2019 ex-post visits to closed projects: Summary Report. John Houston, María José Aramburu and Darline Velghe.

See also a Policy report at

https://www.ecocolife.scot/sites/default/files/LIFE%20Invertebrates%20Platform%20Stirling_Summary%20for%20Policy%20Makers%20_final.pdf

Discussion in lines 280 -292 should also include mention to this new strategy as a positive trend in LIFE funding.

.-In lines 104-106 you mention that "To define the amount of funds allocated to each species for each LIFE project, the budget of each project with multiple species was divided equally among the target species". An interesting detailed research would be to verify in the projects with multiple species, if in the content it is possible to get the amount of money spent for each species in acquiring new Land. In this way you will have data on additional bias towards specific species with particular traits.

-You mention that the species covered by a greater number of LIFE projects were also those which attracted the most interest online (results summarized on Fig. 2b). However, as LIFE projects include many education activities it can happen that those online interest a consequence of the outreach activities of the projects. To have an independent assessment is not an easy task....Why not see if there are peaks of activity towards a given species and remove data from those peaks as biased data!?

-When mentioning the strategies for the future, please also cite the recent paper:

Harvey, J.A., et al. (2020). International scientists formulate a roadmap for insect conservation and recovery. *Nature Ecology and Evolution*, 4: 174–176. DOI:10.1038/s41559-019-1079-8

Review form: Reviewer 2 (Divya Karnad)

Recommendation

Accept with minor revision (please list in comments)

Scientific importance: Is the manuscript an original and important contribution to its field?

Good

General interest: Is the paper of sufficient general interest?

Excellent

Quality of the paper: Is the overall quality of the paper suitable?

Good

Is the length of the paper justified?

Yes

Should the paper be seen by a specialist statistical reviewer?

Yes

Do you have any concerns about statistical analyses in this paper? If so, please specify them explicitly in your report.

No

It is a condition of publication that authors make their supporting data, code and materials available - either as supplementary material or hosted in an external repository. Please rate, if applicable, the supporting data on the following criteria.

Is it accessible?

Yes

Is it clear?

Yes

Is it adequate?

N/A

Do you have any ethical concerns with this paper?

No

Comments to the Author

This manuscript is clearly written and raises important questions about the scientific basis upon which conservation funding is allocated across different species. However, there are certain implicit biases within the manuscript that need to be addressed, before the manuscript can be viewed as identifying the problem objectively.

1) Species-based conservation: This manuscript tackles only this approach to conservation and makes that clear in the methods. However, the reasoning for this choice is not clearly laid out in the introduction. In a manuscript discussing an ideal allocation of limited funding, is species

based conservation the best approach to conservation? Isn't there more emerging research that suggests other approaches, such as ecosystem-based conservation, may offer better returns on investment, so to speak? Could you explain more convincingly why you believe that offering species based approaches to a more diverse set of species will actually produce real-world conservation results that are better than at present?

2) Outcomes of funding: Since this paper looks at over 2 decades of funding, it is assumed that this funding would have had time to produce results in the real world. While it is implied that more funding produces better conservation results, this is not necessarily the case, in my experience. Could you please establish whether the funding helped conserve the species that have been regularly funded over this time? What has this funding bias actually cost the species that are not funded, in terms of their survival and conservation? Would they receive fringe benefits in terms of occupying the same space as some of the larger vertebrates? Establishing these facts would help to build your case.

3) In the discussion you identify that very few invertebrates (in proportion to their diversity) have been assessed for their global conservation status by IUCN. If science is unaware about whether the unassessed species need conservation, it makes sense (to scientists, perhaps) that there would be little funding available for unassessed species. Could you discuss this a bit further, Could you please also report your results in terms of what proportion of IUCN assessed vertebrates and invertebrates receive conservation funding.

Are conservation budgets supposed to cover basic research about newly discovered species and provide the data that might go towards their IUCN assessment? The discussion seems to suggest this, but it needs to be clearly stated. What about transboundary species? What if a heretofore unknown species is sparsely distributed in the EU (which is at the periphery of it's range) but widely distributed in Asia, where conservation budgets don't cover unknown/unassessed species? Would they merit conservation funding?

4) Lines 180 and 181 are repeated.

5) It seems redundant to report that where there is more funding, there are more number of projects. It might be more useful to report whether more funding = more successful conservation outcomes (in whatever way conservation is defined by that funding call).

6) Lines 212-222: It may be useful to bring the gist of this paragraph into the introduction to set the stage for your argument about why IUCN listing may not be relevant to how funding should be distributed across species.

Decision letter (RSPB-2020-2166.R0)

13-Oct-2020

Dear Dr Mammola:

Your manuscript has now been peer reviewed and the reviews have been assessed by an Associate Editor. The reviewers' comments (not including confidential comments to the Editor) and the comments from the Associate Editor are included at the end of this email for your reference. As you will see, the reviewers and the Editors have raised some concerns with your manuscript and we would like to invite you to revise your manuscript to address them.

To submit your revision please log into <http://mc.manuscriptcentral.com/prsb> and enter your Author Centre, where you will find your manuscript title listed under "Manuscripts with

Decisions." Under "Actions", click on "Create a Revision". Your manuscript number has been appended to denote a revision.

Research ethics:

Use of animals and field studies:

It is a condition of publication that you make available the data and research materials supporting the results in the article. Please see our Data Sharing Policies (<https://royalsociety.org/journals/authors/author-guidelines/#data>). Datasets should be deposited in an appropriate publicly available repository and details of the associated accession number, link or DOI to the datasets must be included in the Data Accessibility section of the article (<https://royalsociety.org/journals/ethics-policies/data-sharing-mining/>). Reference(s) to datasets should also be included in the reference list of the article with DOIs (where available).

All supplementary materials accompanying an accepted article will be treated as in their final form. They will be published alongside the paper on the journal website and posted on the online figshare repository. Files on figshare will be made available approximately one week before the

accompanying article so that the supplementary material can be attributed a unique DOI. Please try to submit all supplementary material as a single file.

Please submit a copy of your revised paper within three weeks. If we do not hear from you within this time your manuscript will be rejected. If you are unable to meet this deadline please let us know as soon as possible, as we may be able to grant a short extension.

Best wishes,

Professor Gary Carvalho
mailto:proceedingsb@royalsociety.org

Associate Editor
Board Member: 1
Comments to Author:

This is an interesting and important analyses, because conservation investment appears to be skewed. The authors find very interesting species bias for conservation funds in a large dataset from the EU. While such investigations are essential and the insights uncovered by the analyses fascinating, referee 2 raises some important points about outcomes of funding and other details such as species status and details of proposed research. The authors are requested to revise their manuscript as suggested by the referees. Following such revision, this maybe a good manuscript for proceedings b.

Reviewer(s)' Comments to Author:

Referee: 1

Comments to the Author(s)

It is well known that there is a bias in the funds allocated to the conservation of species with a clear positive bias towards vertebrates. Invertebrates were clearly overlooked. This manuscript reports on an interesting evaluation of the taxonomic biases on LIFE projects conducted between 1992 and 2018, and potential drivers and their possible impact on conservation strategies and decision-making policies. The research was well conducted and I only suggest some minor improvements.

Some comments for improvement:

-In the Introduction, you should also mention that LIFE program is now targeting LIFE projects focusing on Invertebrates and refer to:

Neemo (2020) LIFE and Invertebrates-Stepping up to the challenges: Conclusions of the 2018 LIFE Platform Meeting on invertebrates and 2019 ex-post visits to closed projects: Summary Report. John Houston, María José Aramburu and Darline Velghe.

See also a Policy report at

https://www.ecocolife.scot/sites/default/files/LIFE%20Invertebrates%20Platform%20Stirling_Summary%20for%20Policy%20Makers%20_final.pdf

Discussion in lines 280 -292 should also include mention to this new strategy as a positive trend in LIFE funding.

-In lines 104-106 you mention that “To define the amount of funds allocated to each species for each LIFE project, the budget of each project with multiple species was divided equally among the target species”. An interesting detailed research would be to verify in the projects with multiple species, if in the content it is possible to get the amount of money spent for each species in acquiring new Land. In this way you will have data on additional bias towards specific species with particular traits.

-You mention that the species covered by a greater number of LIFE projects were also those which attracted the most interest online (results summarized on Fig. 2b). However, as LIFE projects include many education activities it can happen that those online interest a consequence of the outreach activities of the projects. To have an independent assessment is not an easy task....Why not see if there are peaks of activity towards a given species and remove data from those peaks as biased data!?

-When mentioning the strategies for the future, please also cite the recent paper:

Harvey, J.A., et al. (2020). International scientists formulate a roadmap for insect conservation and recovery. *Nature Ecology and Evolution*, 4: 174-176. DOI:10.1038/s41559-019-1079-8

Referee: 2

Comments to the Author(s)

This manuscript is clearly written and raises important questions about the scientific basis upon which conservation funding is allocated across different species. However, there are certain implicit biases within the manuscript that need to be addressed, before the manuscript can be viewed as identifying the problem objectively.

1) Species-based conservation: This manuscript tackles only this approach to conservation and makes that clear in the methods. However, the reasoning for this choice is not clearly laid out in the introduction. In a manuscript discussing an ideal allocation of limited funding, is species based conservation the best approach to conservation? Isn't there more emerging research that suggests other approaches, such as ecosystem-based conservation, may offer better returns on investment, so to speak? Could you explain more convincingly why you believe that offering species based approaches to a more diverse set of species will actually produce real-world conservation results that are better than at present?

2) Outcomes of funding: Since this paper looks at over 2 decades of funding, it is assumed that this funding would have had time to produce results in the real world. While it is implied that more funding produces better conservation results, this is not necessarily the case, in my experience. Could you please establish whether the funding helped conserve the species that have been regularly funded over this time? What has this funding bias actually cost the species that are not funded, in terms of their survival and conservation? Would they receive fringe benefits in terms of occupying the same space as some of the larger vertebrates? Establishing these facts would help to build your case.

3) In the discussion you identify that very few invertebrates (in proportion to their diversity) have been assessed for their global conservation status by IUCN. If science is unaware about whether the unassessed species need conservation, it makes sense (to scientists, perhaps) that there would be little funding available for unassessed species. Could you discuss this a bit further, Could you please also report your results in terms of what proportion of IUCN assessed vertebrates and invertebrates receive conservation funding.

Are conservation budgets supposed to cover basic research about newly discovered species and provide the data that might go towards their IUCN assessment? The discussion seems to suggest this, but it needs to be clearly stated. What about transboundary species? What if a heretofore

unknown species is sparsely distributed in the EU (which is at the periphery of it's range) but widely distributed in Asia, where conservation budgets don't cover unknown/unassessed species? Would they merit conservation funding?

4) Lines 180 and 181 are repeated.

5) It seems redundant to report that where there is more funding, there are more number of projects. It might be more useful to report whether more funding = more successful conservation outcomes (in whatever way conservation is defined by that funding call).

6) Lines 212-222: It may be useful to bring the gist of this paragraph into the introduction to set the stage for your argument about why IUCN listing may not be relevant to how funding should be distributed across species.

Author's Response to Decision Letter for (RSPB-2020-2166.R0)

See Appendix A.

RSPB-2020-2166.R1 (Revision)

Review form: Reviewer 1 (Divya Karnad)

Recommendation

Accept as is

Scientific importance: Is the manuscript an original and important contribution to its field?

Good

General interest: Is the paper of sufficient general interest?

Good

Quality of the paper: Is the overall quality of the paper suitable?

Good

Is the length of the paper justified?

Yes

Should the paper be seen by a specialist statistical reviewer?

Yes

Do you have any concerns about statistical analyses in this paper? If so, please specify them explicitly in your report.

No

It is a condition of publication that authors make their supporting data, code and materials available - either as supplementary material or hosted in an external repository. Please rate, if applicable, the supporting data on the following criteria.

Is it accessible?

Yes

Is it clear?

Yes

Is it adequate?

Yes

Do you have any ethical concerns with this paper?

No

Comments to the Author

No further comments.

Decision letter (RSPB-2020-2166.R1)

17-Nov-2020

Dear Dr Mammola

I am pleased to inform you that your manuscript entitled "Towards a taxonomically unbiased EU Biodiversity Strategy for 2030" has been accepted for publication in Proceedings B.

Open Access

Your article has been estimated as being 7 pages long. Our Production Office will be able to confirm the exact length at proof stage.

Paper charges

Sincerely,

Professor Gary Carvalho
Editor, Proceedings B
mailto: proceedingsb@royalsociety.org

Associate Editor:

Board Member: 1

Comments to Author:

The authors have made the required revisions. This paper is now acceptable for publication in Proceedings B.

Board Member: 2

Comments to Author:

Referee 2 brought up several important points that the authors have addressed, but it would be best to ensure that these revisions are adequate. Review of the revisions by Referee 2 is adequate.

Appendix A

Point-by-point responses:

Associate Editor - Board Member: 1

This is an interesting and important analysis, because conservation investment appears to be skewed. The authors find very interesting species bias for conservation funds in a large dataset from the EU. While such investigations are essential and the insights uncovered by the analyses fascinating, referee 2 raises some important points about outcomes of funding and other details such as species status and details of proposed research. The authors are requested to revise their manuscript as suggested by the referees. Following such revision, this may be a good manuscript for proceedings b.

Response: Thank you for handling this submission so efficiently, for your positive feedback, and the useful editorial guidance on the direction to give to this revision. As detailed below, we have taken into consideration all comments by the two referees.

>> Referee: 1

Comments to the Author(s)

It is well known that there is a bias in the funds allocated to the conservation of species with a clear positive bias towards vertebrates. Invertebrates were clearly overlooked. This manuscript reports on an interesting evaluation of the taxonomic biases on LIFE projects conducted between 1992 and 2018, and potential drivers and their possible impact on conservation strategies and decision-making policies. The research was well conducted and I only suggest some minor improvements.

Response: Thank you for taking the time to carefully review our work and your positive and constructive feedback.

Some comments for improvement:

-In the Introduction, you should also mention that LIFE program is now targeting LIFE projects focusing on Invertebrates and refer to:

Neemo (2020) LIFE and Invertebrates-Stepping up to the challenges: Conclusions of the 2018 LIFE Platform Meeting on invertebrates and 2019 ex-post visits to closed projects: Summary Report. John Houston, María José Aramburu and Darline Velghe.

See also a Policy report at

<https://www.ecocolife.scot/>

Discussion in lines 280 -292 should also include mention to this new strategy as a positive trend in LIFE funding.

Response: When drafting the paper we overlooked this important document. We have now cited this source in both the introduction and the discussion.

-In lines 104-106 you mention that “To define the amount of funds allocated to each species for each LIFE project, the budget of each project with multiple species was divided equally among the target species”. An interesting detailed research would be to verify in the projects with multiple species, if in the content it is possible to get the amount of money spent for each species in acquiring new Land. In this way you will have data on additional bias towards specific species with particular traits.

Response: This is a great suggestion for an interesting research that would, however, require in-depth analysis aimed at mining information on the different usage of conservation funds in the frame of each LIFE project. However, we felt it was outside the scope of this short contribution and of the time-frame provided for revising the manuscript: it is extremely hard to gather such data as most LIFE projects focusing on land acquisition and recovery are usually targeting multiple species without mentioning individual contributions.

-You mention that the species covered by a greater number of LIFE projects were also those which attracted the most interest online (results summarized on Fig. 2b). However, as LIFE projects include many education activities it can happen that those online interest a consequence of the outreach activities of the projects. To have an independent assessment is not an easy task...Why not see if there are peaks of activity towards a given species and remove data from those peaks as biased data!?

Response: Thanks for pointing out this. We probably did not explain very clearly the nature of our “popularity” metric, thus generating a fertile ground for potential misunderstanding.

Popularity represents the average of monthly search volume for a species across 10 years of data (120 months of search volume). Because of this, and while there is a possibility that there are occasional spikes in interest, the "signal" in our data is more likely to represent the underlying degree of public interest in each species over the 10 years that were sampled. Another characteristic of the metric is that it is a relative metric that is scaled across species, so the volume of searches can be compared between species in proportional terms. To give a bit of context, and based on our data, the Wolf (most searched species) has about 10 times more the number of monthly searches for Brown bear and about 125 times (*Sic!*) more the number of monthly searches for stag beetles, for example. For the least popular species, the difference is even more abysmal.

Furthermore, we visually inspected the data for the most frequently searched species and there were no individual clear spikes in search volume. Such spikes may be more likely to be found in less popular species that are the target of sudden peaks of interest (e.g. in association with outreach activities), but removing these spikes from the data will only emphasize the differences to the most popular species and reinforce the stark contrast in popularity between species. We thus believe that having such pronounced differences between species on a monthly basis over 10 years of data provides strong grounds to argue that some species are consistently more searched for on the internet, and thus more popular than others.

To avoid any source of potential confusion for readers, we have now further clarified the scope of this metric in the methods (Line 160–164).

-When mentioning the strategies for the future, please also cite the recent paper:

Harvey, J.A., et al. (2020). International scientists formulate a roadmap for insect conservation and recovery. *Nature Ecology and Evolution*, 4: 174–176. DOI:10.1038/s41559-019-1079-8

Response: Thanks, done.

>> Referee: 2

Comments to the Author(s)

This manuscript is clearly written and raises important questions about the scientific basis upon which conservation funding is allocated across different species. However, there are certain implicit biases within the manuscript that need to be addressed, before the manuscript can be viewed as identifying the problem objectively.

Response: Thank you for spending time in reviewing our work, your positive attitude towards it, and for introducing these useful points of discussion.

1) Species-based conservation: This manuscript tackles only this approach to conservation and makes that clear in the methods. However, the reasoning for this choice is not clearly laid out in the introduction. In a manuscript discussing an ideal allocation of limited funding, is species based conservation the best approach to conservation? Isn't there more emerging research that suggests other approaches, such as ecosystem-based conservation, may offer better returns on investment, so to speak? Could you explain more convincingly why you believe that offering

species based approaches to a more diverse set of species will actually produce real-world conservation results that are better than at present?

Response: Yes, we fully agree that a species-based approach may not be the most suited conservation strategy, especially for mega-diverse groups such as arthropods. Please consider, however, that our aim here is not to discuss the effectiveness of different conservation strategies (e.g., actions toward single species vs the use of general protected areas vs habitat/ecosystem or even socio-ecological focus). Rather, we seek to explore the usage of budget in the frame of existing conservation initiatives in Europe (i.e. LIFE).

Through the course of the Habitats Directive, the EU adopted the strategy of spending a significant fraction of the total LIFE budget directly on species-level conservation actions. Even though this may not be the most effective conservation strategy, it was the dominant approach for conservation funding allocation. Based on this *status quo*, we believe it is important to emphasize the huge bias in the allocation of funds for species-focused conservation.

Likewise, in the future Biodiversity Strategy for 2030, the EU will once again spend a significant proportion of budget to ensure “... *no deterioration in conservation trends and the status of protected species*”.

In this scenario, our work seeks to ensure that a similarly skewed investment of funds will not happen again. We now included more details in the introduction (Line 92–93) and methods (Line 107–109) to better delineate the boundaries of our analysis, and we also mention alternative conservation mechanisms where relevant in the recommendations.

2) Outcomes of funding: Since this paper looks at over 2 decades of funding, it is assumed that this funding would have had time to produce results in the real world. While it is implied that more funding produces better conservation results, this is not necessarily the case, in my experience. Could you please establish whether the funding helped conserve the species that have been regularly funded over this time? What has this funding bias actually cost the species that are not funded, in terms of their survival and conservation?

Response: These are all very interesting questions. On the one hand, there is published evidence that some species that received constant funds in the past (wolf, brown bear, lynx) are now in better conservation status (Chapron et al. 2014). In the same vein, a recent study by Rosenberg et al. (2019) showed that the only group of North American birds with an improved status, when compared to the 1970s, were the one inhabiting wetlands, which is the habitat that attracted the most funds. On the other hand, we are afraid that we lack the data to try answering the second question about the downside of

not being funded. As a consequence of the taxonomic and other biases, we simply know too little about the status of these species that are neglected in conservation policies. We now incorporated these arguments in the introduction (Lines 63–69).

Chapron, G. et al. 2014. Recovery of large carnivores in Europe's modern human-dominated landscapes. *Science* 346: 1517–1519.

Rosenberg, K. V., Dokter, A. M., Blancher, P. J., Sauer, J. R., Smith, A. C., Smith, P. A., ... & Marra, P. P. (2019). Decline of the North American avifauna. *Science*, 366(6461), 120-124

Would they receive fringe benefits in terms of occupying the same space as some of the larger vertebrates?

Response: This is a somewhat controversial topic, making it difficult to provide a univocal answer. On the one hand, evidence suggests that sometimes occupying the same habitat of a large vertebrate (i.e., an umbrella species) can provide conservation benefits for certain species (Caro 2010)—we have briefly touched this issue at Lines 238–243. On the other hand, there is counter-evidence showing the opposite (or no) pattern for other species (e.g., Stewart et al., 2018; Vessby et al., 2002). The main issue here is that even if the distribution range of a brown bear and a beetle seemingly overlap, their niche may not. Most invertebrates simply operate at a different scale than vertebrates, and their conservation needs can be drastically different as a result.

Caro, T. (2010). *Conservation by proxy: indicator, umbrella, keystone, flagship, and other surrogate species*. Island Press. - This book is perfect for what we want... and the next two papers are also good examples:

Stewart, D. R., Underwood, Z. E., Rahel, F. J., & Walters, A. W. (2018). The effectiveness of surrogate taxa to conserve freshwater biodiversity. *Conservation Biology*, 32(1), 183-194.

Vessby, K., Söderström, B. O., Glimskär, A., & Svensson, B. (2002). Species–richness correlations of six different taxa in Swedish seminatural grasslands. *Conservation Biology*, 16(2), 430-439.

3) In the discussion you identify that very few invertebrates (in proportion to their diversity) have been assessed for their global conservation status by IUCN. If science is unaware about whether the unassessed species need conservation, it makes sense (to scientists, perhaps) that there would be little funding available for unassessed species. Could you discuss this a bit further? Are conservation budgets supposed to cover basic research about newly discovered species and provide the data that might go towards their IUCN assessment? The discussion seems to suggest this, but it needs to be clearly stated. What about transboundary species? What if a heretofore unknown species is sparsely distributed in the EU (which is at the periphery of its range) but widely distributed in Asia, where conservation budgets don't cover unknown/unassessed species? Would they merit conservation funding?

Response: In principle, the LIFE projects do not cover basic research, crucial to feed proper conservation status (IUCN) assessments, but only applied conservation measures on a limited number of species that are listed in the Annexes of the Habitat Directive. Practically speaking, species that are not listed in the habitat directive are simply not targeted by LIFE projects (although during the later years species assessed as threatened in the global IUCN Red List can be used to justify specific conservation actions). These include most of the potential cases that the referee is mentioning, such as the transboundary Asian species. The problem is that, in turn, the habitat directive is biased in its coverage toward vertebrates and just a few charismatic invertebrate species (butterflies, some large sized beetles, and not much more), generating a vicious circle. We now mentioned the need of revising the annexes to include a more objective coverage of species across the Tree of Life in the recommendations.

4) Lines 180 and 181 are repeated.

Response: Removed.

5) It seems redundant to report that where there is more funding, there are more number of projects. It might be more useful to report whether more funding = more successful conservation outcomes (in whatever way conservation is defined by that funding call).

Response: Actually, we believe that this information is not entirely redundant, as it highlights different approaches on how the budget on conservation is spent by the EU. We discussed it as follows:

As expected, we found a significant positive correlation between budget allocation and the number of projects (Fig. 2a), although with key differences in conservation strategies for distinct species. Whereas the majority of species (including all invertebrates) were covered by a small number of projects with a limited budget (€0–5 million), some outlying species were the target of more intense budget allocation and a higher number of projects (e.g., the bear and great bittern), or a lower number of high-budget projects (e.g., the lynx and wolf).

We would prefer to maintain this information, but of course we are fully open to reconsider our position if deemed appropriate.

6) Lines 212-222: It may be useful to bring the gist of this paragraph into the introduction to set

the stage for your argument about why IUCN listing may not be relevant to how funding should be distributed across species.

Response: Good point. Done at lines 83–88.